# Mesh Graft Urethroplasty—Still a Safe and Promising Technique in Mostly Unpromising Complex Urethral Strictures

**DOI:** 10.3390/jcm11205989

**Published:** 2022-10-11

**Authors:** Mathias Reichert, Maurizio Aragona, Ahmad Soukkar, Roberto Olianas

**Affiliations:** 1Department of Urology, Universitätsmedizin Göttingen, 37099 Göttingen, Germany; 2Department of Urology, Städtisches Klinikum Lüneburg, 21339 Lüneburg, Germany

**Keywords:** two-staged urethroplasty, complex urethral strictures, spongiofibrosis, mesh graft urethroplasty

## Abstract

Long urethral strictures or even recurrent urethral strictures, mostly with scar tissue showing insufficient healing tendencies, are defined as complex and represent a big challenge in modern reconstructive urology. Initially, the treatment of complicated urethral strictures was associated with a high failure rate (20–40%) due to the growth of hair in the neourethra and a lack of sufficient suitable epithelium when scrotal skin was used. Although much effort was put into tissue engineering recently, harvesting and transplanting autologous tissue represent the standard of care for urethral substitution or augmentation. Since 1977, two-staged urethroplasty with the usage of free foreskin or 0.1 mm thick meshed skin from the upper leg was performed in complicated cases and was initially described in 1984 and 1989 by Schreiter and Schreiter and Noll, respectively. In stage 1, the graft is harvested by cutting the skin thinly above the hair follicles and transplanted as a plate around the opened urethra. In stage 2, after 8–12 weeks, the neourethra is formed. Success rates of up to 84% are described. Considering the complexity of the strictures in which mesh graft urethroplasty is usually performed, the reachable success rates are outstanding, especially considering that this surgery is most likely the last opportunity to prevent perineostomy or even urinary diversion. This article describes the surgical technique and embeds the mesh graft urethroplasty in today’s literature to underline its importance in the surgical management of complex urethral strictures.

## 1. Introduction

The use of different surgical approaches for different urethral strictures is obligatory for the success of the treatment.

Depending on stricture localization and its length, as well as on the degree of the accompanying spongiofibrosis, different techniques may be used, whereby spongiofibrosis is defined as hyperpöastical structural change in the corpus spongiosum tissue, mainly because of scar tissue. Spongiofibrosis is categorized in different degrees of severity. Minimal spongiofibrosis is characterized as scarring tissue just around the urethral mucosa, moderate to complete spongiofibrosis affects the complete corpus spongiosum that surrounds the strictured urethra (the spongiofibrosis can extend further than the actual stricture). The next stage is defined as spongiofibrosis, which affects tissue next to the corpus spongiosum accompanied by inflammation, followed by complex spongiofibrosis with fistula development, which can lead to an abscess formation [1].

The etiology of the stricture and previous history of urethral surgery may influence the choice of the surgical procedure [2].

If there is a residual lumen of the urethra and the severity of the spongiofibrosis is manageable (mild to moderate), augmentation urethroplasty seems to be a therapeutical option. This can be performed in one- or two-staged surgery.

Complex urethral strictures remain a challenging situation for reconstructive urologic surgeons. These are defined as long and recurrent strictures, mainly after failed urethral stricture surgery, with bad tissue conditions that are associated with bad healing tendencies.

Therefore, these strictures are often characterized by poor success rates and often by the need for reintervention [3].

Completely obliterated urethras with a length such that a primary resection and the re-anastomosis of the urethra are not an option, caused by scar tissue out of chronic inflammation and often combined with abscesses and stones, should be treated in a two-staged procedure to minimize complications caused by the intervention itself.

Two-staged procedures started with the classic use of scrotal skin, described by Johanson in 1953 [4]. Unfortunately, this surgical technique led to a 30% failure rate in long-term follow-up, mainly because of the growth of hair and the consequent inflammation in the neorurethra.

Therefore, Schreiter and Noll started to use a different type of autologous tissue for grafting into the strictured urethra: meshed split thickness skin or foreskin [5]. With this technique, they could provide a large area of hairless epithelium that can be reconstructed in a neourethra in a second stage. Especially in circumcised patients, 0.1 mm thin split-skin graft harvested from the upper leg is feasible providing an appropriate graft without hair follicles [3].

The initial tests showed satisfactory results with 98.71% success even after follow-up up to 8 years. In 14.10% of the patients, smaller re-interventions like meatotomy had to be performed [3]. These results supported the role of this procedure in the treatment of complex strictures where urinary diversion could potentially be the only alternative surgical intervention.

## 2. Surgical Technique

The surgical procedure consists of two stages.

### 2.1. First Stage

The surgical approach for the first-stage MG urethroplasty is the same as any other urethroplasty with a perineal approach. If the stricture is over the entire length of the urethra, the scrotum has to be divided at the raphe. After the urethra is disposed, a metal bougie is put retrograde into the urethra to identify the beginning of the stricture, and afterwards, the urethra is opened at the tip of the metal bougie. Now, the urethra has to be opened longitudinal into healthy tissue (see Figure 1). The divided scrotum is sutured laterally in a horizontal manner over the testes (see Figure 2).

Before removing all scar tissue to prepare the recipient bed for the skin transplant, the length of the opened urethra has to be measured.

A split-skin dermatome is used to harvest the graft from the inner side of the thigh. The harvest-area has to be healthy with no scars (see Figure 3). It is important to decide together with the patient preoperatively which leg should be used.

The skin transplant has to be cut thinly enough to ensure that no hair follicles remain in the harvest. Usually a thickness of 0.1–0.2 mm is sufficient. No tendency of curling of the transplant and semitransparency are intraoperative signs that the thickness seems to be correct. After meshing the split-skin transplant with a dermatome, the surface enlarges about 50–100%, and this should guarantee a surface of about 100–120 cm^2^ (see Figure 4).

The meshed skin graft has several advantages besides gaining more tissue surface. It provides optimal adaption of the harvest on the mainly uneven recipient bed and optimal drainage conditions for wound secretion and smaller bleedings. This circumstance seems to be the most important reason for successful healing in the first postoperative phase because it guarantees a good neovascularization of the transplant. In case of the retention of wound secretions and blood, a neovascularization cannot occur in the right way and results in the shrinkage and restructuring of the neourethra [4]. This can often be observed in closed full-thickness skin patches, which are only perforated instead of meshed [6,7].

The harvested and meshed transplant is tailored to the needed size and sutured onto the prepared recipient bed between the urethral plate and opened skin edge (5-0 interrupted running monofil resorbable sutures) (see Figure 5).

Because of the already described advantage of an ideal adaptation to the irregular underlying tissue, the transplant achieves the required contact for a rapid neovascularization and the perfect healing of the graft.

After completion of the sutures a fibrin glue is applied in order to improve the adhesion of the skin on the transplantation-bed. Paraffin gauze is applied onto the transplant (see Figure 6). The complete wound is covered by a loosely applied compression dressing for seven days (see Figure 6).

### 2.2. Postoperative Management

Postoperatively, even small movements of the transplant can cause a delay in the engraftment of the transplant. Therefore, the patient has to stay in bed for seven days as well as maintaining the compressive dressing. During that time, bowel activity has to be reduced using appropriate drugs (morphine and loperamide), and the patients receive parenteral nutrition.

Urinary drainage is achieved by suprapubic cystotomy.

From the seventh day on, daily changing of the dressing guarantees assessing of the healing process. While changing the dressing, care must be taken on preventing the transplant form lifting of from its bed. In case of hematoma, the skin should be incised, so that the graft has a good adaption to the underlying.

To prevent scar tissue formation and skin bridges in the area of the transplant surface, a loose paraffin gauze dressing is applied between the transplant surfaces after every change of dressing. If a skin bridge developed, it had to be incised.

After the sufficient healing of the transplant, patients should take a daily chamomile-solution bath. The suprapubic catheter can be removed if the transplant is sufficiently healed and micturition via the perineostomy is sufficient.

### 2.3. Second Stage

The second stage of MG urethroplasty, with the reconstruction of the urethra from the completely healed transplant, should not be undertaken before a period of 3 months has elapsed.

After that period of time, it is guaranteed that the “new” urethral plate is completely epitheilialized and neovascularization is finished. By then, the transplant should ensure optimal conditions for the reconstruction of the neourethra.

During the preoperative evaluation before the second-stage surgery, an accurate inspection of the skin should be made in order to assess the compete healing, exclude retraction spots and assess the caliber of the proximal urethra with a 22CH transurethral catheter (Figure 7). In case of retraction of the skin, this retraction can be corrected during the second-stage procedure. Detention incisions, flap of the grafted skin or even augmentation with buccal mucosa may help in these cases.

For the first step, the transplant is incised longitudinally in an appropriate distance to the original urethral plate to ensure a sufficient wide neourethra after closing (Figure 8). At the proximal and distal endings of the original urethral plate, the incision should be performed closer to the urethra, so that there would not be a urethral diverticulum after closing it.

To close the neourethra, the lateral incisions are sutured over a 24 FG catheter by interrupted running sutures. To prevent fistula formation these sutures should be inverted by not penetrating the epidermis (Figure 9).

The laterally fixed scrotal skin during the first procedure is dissected again, and the normal anatomy is reconstructed by approximating the two scrotal skin halves in the midline to reproduce a raphe (Figure 10).

The penile skin is closed using a Byars/Marberger flap technique [8,9]. Before that, the meatal edges of the glans penis have to be trimmed.

After completing the closing the urethral catheter is removed, and a light dressing is applied.

### 2.4. Postoperative Management

Similar to the first-stage procedure, a suprapubic cystotomy is applied for urinary drainage. After urethral catheter removal at the end of the second-stage procedure, a tube is placed into the urethra to drain urethral secretion. Five days after surgery, the light compression dressing is removed.

Ten days after the second-stage surgery, a micturition cystourethrogram is performed via the suprapubic cystotomy, which is removed after proof of watertight urethral sutures.

## 3. Discussion

Urethral strictures have to be diagnosed and classified the right way so that the ideal surgical technique can be found and be discussed with the patient, but even if the correct surgical techniques to be used are preoperatively chosen, surgeons have to be flexible during the operation since intraoperative findings can potentially lead to the necessity of changing the surgical approach.

To be able to classify urethral strictures in a proper way, the etiology and pathologic progression of strictures have to be understood. Urethral strictures mostly develop out of the cicatricial contraction of the corpus spongiosum, which is defined as spongiofibrosis.

The severity of spongiofibrosis is heterogeneous and depends on its etiology (inflammation, trauma, etc.). It mostly occurs out of an infection of the urethra caused by indwelling catheter use with a secretion of mucous in conjunction with pressure damage caused by the catheter. This leads to the formation of periurethral infiltrates, which leads to the formation of scarred bridges between opposite regions of mucous membranes [10].

There are histological and immunohistochemical changes that can be observed in the scarring corpus spongiosum, surrounding the strictured urethra. Baskin et al. could identify collagen and bundles of elastin packed in around the thickened epithelial layer of the strictured urethral wall [11].

Spongiofibrosis, with it scar tissue leading to urethral strictures, might be a result of inflammatory, traumatic, ischemic, congenital or iatrogenic factors [12]. The ability to successfully treat urethral strictures has increased in recent decades because of the possibilities of more surgery via the urethra (endoscopic transurethral) [13].

The extent of spongiofibrosis is critical for deciding which kind of surgical therapy should be used since the “urethral repair” should be performed into healthy tissue. Therefore, the first challenge for surgeons is to diagnose the true length of the damaged tissue.

The most commonly used methods of evaluating the extent, number and localization of urethral strictures are retrograde urethrography (RUG) and voiding cystourethrography (VCUG). These also enable the diagnosis of urethral wall damage, fistula and false passage [14]. However, RUG and VCUG are not able to show pathologies of the urethra beyond the strictured lumen. In addition, the significant radiation exposure can be a problem, especially in younger patients with expected numerous re-examinations [15]. Nevertheless, it is an easy-to-perform clinical tool for providing an idea of the pathological changes. In Germany, RUG and VCUG are urologist-performed. This seems to be an important issue since studies showed that the preoperative assessment evaluated by radiologists showed correctness in only 87% of cases, compared with 96% when assessed by urologists [16]. Furthermore, only 49% of the reports generated by radiologists seem to be adequate for interpretation [16].

Another important tool in diagnosing urethral strictures and spongiofibrosis with easy everyday access is sonourethrography (SUG). Choudhary et al. identified that SUG as more accurate in the bulbar urethra than it is by RUG/VCUG in evaluating the length of the stricture and enabled assessing the corpus spongiosum [17], especially in strictures arising from traumatic catheterization and idiopathic stenoses. The disadvantages of this technique, however, are its high subjectivity, the risk of over-diagnosing of strictures and the impossibility of evaluating long or multiple strictures [18].

Magneturethrourethrography (MRU) seems to overcome the disadvantages of RUG/VCUG and SUG. Numerous publications had been focusing on MRU in recent years. MRU provides good soft tissue contrast for determining the degree and extent of spongiofibrosis [19]. The objectiveness of the procedure, absence of radiation exposure and provision of additional information on the surrounding organs and tissues after pelvic trauma are among its strengths [20]. However, it is not a very cost-effective method, it is not accessible for every urologist and it necessitates a dependency on radiologists.

In our opinion, RUG/VCUG are still the gold standard in diagnosing and characterizing urethral strictures. Access to the posterior urethra requires mostly a combined ante- and retrograde urethrography [21]. These procedures, combined with carefully performed cystoscopy (ante- and retrograde cystoscopy via suprapubic access), offer all information needed for experienced urologists to provide the best surgical care for affected patients. In rare cases, extended diagnostic tools could find their place (e.g., MRU for pelvic traumas).

To date, urologists have attempted to find the best surgical methods for complex urethral strictures—more precisely, long urethral strictures with severe spongiofibrosis because of scar tissue affecting the urethral wall—since the attendant burdens that come with the unhealthy tissue surrounding the strictured urethra make a free graft transplantation difficult.

Complete panurethral defects of the urethra normally require a two-staged process. This is one out of the three important principles for successfully performing surgery in complex panurethral strictures besides loose, tension-free transplantation of skin flaps and usage of hairless skin. Using a free buccal mucosa graft in these situations entails some limitations and dangers, even when performed in two stages. To successfully transplant buccal mucosa in these patients, a healthy and well-vascularized urethral plate has to be formed. This is usually reached with a fascia dartos flap [10]. After a successful reconstruction of the neourethra out of BM, there is still the risk of developing a penile curvature while healing because of the fibrotic remodeling of the urethral plate if the graft bed is not adequate [10].

In extreme cases with a high degree of spongiofibrosis of the bulbar urethra, a single-staged perienostomy using proximal MG urethroplasty can be performed, especially if the patient declines a two-staged procedure. The use of MG in these cases ensures that a retraction with the risk of stenosis (because of the severe spongiofibrosis) will be unlikely.

Cases are described with two separate strictures in different locations of the urethra (e.g., bulbar and penile) that were successfully repaired using buccal mucosa in both locations [22]. Those cases are limited and should be seen as two separate strictures, and each should provide the basis needed for successfully using BM. Even in long urethral penile strictures, BM is optimal tissue for stricture repair. These strictures usually are combined with spongiofibrosis.

In selected cases, we perform a combination of MG and BM urethroplasty whereby the proximal urethra is supplied with MG and the penile stricture with inlay BM.

Even if the urethral tissue allows for the transplantation of BM, there are situations that force the surgeon not to use it. The harvesting of BM itself can be a problem, for example in cases of prior buccal harvest, heavy tobacco use, oncological disease or oral radiation in the past [23].

To overcome the described disadvantages, tissue engineering found its way into the urological field. Urotiss provided the first tissue engineering product, called mukocell, whereby oral mucosa cells are gained by biopsy and isolated, expanded and cultivated on the surface of a biological matrix. Within 3 weeks, the tissue “grows” in an aseptic laboratory [24]. In cases when we used tissue-engineered harvests, we saw similar outcomes to conventional BM surgeries with a success rate of 84%. Our results embed into the present literature. Studies so far have concluded that tissue-engineered oral mucosa grafts seem to guarantee a similar success rate when using the same surgical techniques for native oral mucosa [25].

Tissue engineering overcomes the limitations at the harvest site but cannot overcome the limitations that accompany the scarred tissue at the recipient site.

Traumatic prostatomembranous distraction of the urethra can be an indication for using mesh graft in the former bulbar urethral region.

Although Gelman and Wisenbough postulate that for prostatomembranous distraction with a complete traumatic transection of the urethra, a primary reanastomosis of the urethra in combination with a (infra-)pubectomy is an adequate way to treat those injuries [21], the use of posterior mesh in those cases represents a good surgical technique for forming a neourethra to bridge the distance of the transected urethra without pubectomy [26]. This can be helpful in complex bone situations.

In conclusion, with regard to the issues described above, there are different limitations in different indications for using BM as free grafts in urethral repair that require a Plan B. In our opinion, using mesh graft for sanitizing complex urethral strictures is not only a “plan B” but should be required whenever there could be an issue for not using BM.

For us, the indications for using two-staged mesh graft urethroplasty are complex strictures, defined as followed:-Multiple strictures.-Recurrent cases.-Wide scar tissue.-Long strictures.

The etiology of stricture does not have a particular decision-making effect for choosing the mesh-graft technique, but of course the attendant circumstances have to be noticed and have some influence.

Between 2006 to 2012, we performed 38 MG urethroplasties as described above, with a success rate of 81.6% (*n* = 31). Stricture length was 4–15 cm. Non-successful patients showed a stricture recurrence in 57%, fistulas in 28.6% and urethral diverticula development in 14.3%.

This is concordant with the results published before [3,5] and shows similar success rates to native urethroplasties with BM augmentation or end-to-end anastomosis.

Pfalzgraf et al. published their results for MG urethroplasty in 2010. They reached recurrence-free urethral repair for MG in 84% of 68 performed MG urethroplasties [2]. In 1998, Schreiter described his results with the use of two-staged urethroplasty with free skin graft and showed an overall success rate of 92% with 63 patients [27].

Regarding of the recurrence risk, Vashishtha et al. described the etiology (post-traumatic) and the degree of fibrosis of stricture length as statistic significant predictors of a potential recurrence for strictures in pediatric patients [28].

In 2010, Pfalzgraf et al. published an interesting study in which patients satisfaction after urethroplasty was evaluated [2]. They compared BM urethroplasty with mesh graft urethroplasty, and satisfaction was reached in 96.7% of cases for BM and 83.3% for MG. However, when the two-staged procedures were compared, satisfaction in the BM group reached 80%, less than for MG.

In conclusion, satisfaction can be reached performing mesh graft. Realistic preoperative patient counseling can improve postoperative subjective outcomes. Patients’ consideration of the outcomes can differ from physicians’ [29], but with realistic expectations, most patients reach postoperative satisfaction.

## 4. Conclusions

Since 1977, two-staged MG urethroplasty has been performed for complex urethral strictures when the most severe spongiofibrosis forces the surgeon to go beyond the common urethroplasty techniques to achieve adequate success. To date, MG urethroplasty represents a safe and promising surgical technique for unpromising urethral circumstances with excellent results that can be reached with corresponding patient satisfaction.

## Figures and Tables

**Figure 1 jcm-11-05989-f001:**
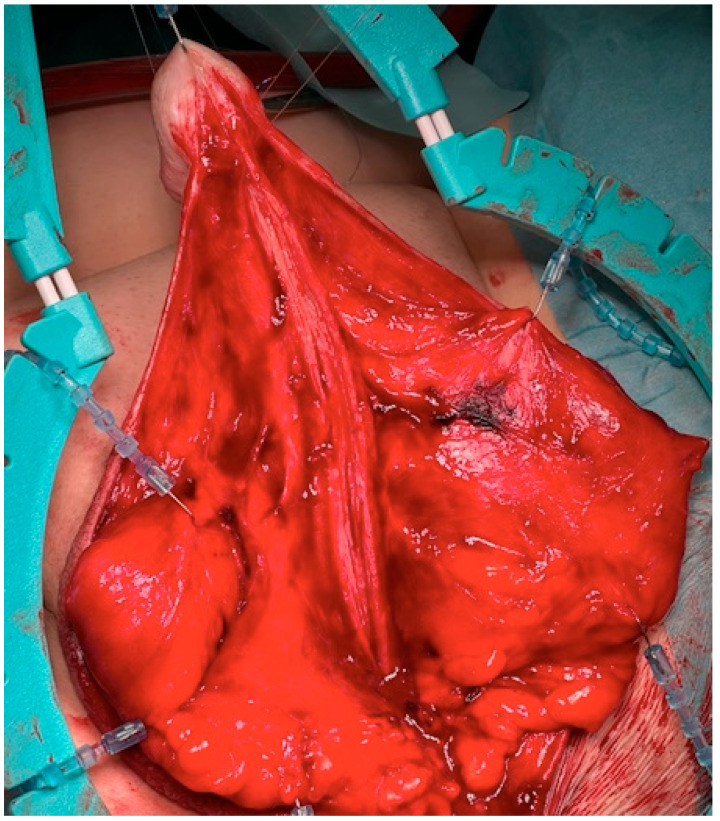
The stricturated urethra is opened and the entire stricture is split.

**Figure 2 jcm-11-05989-f002:**
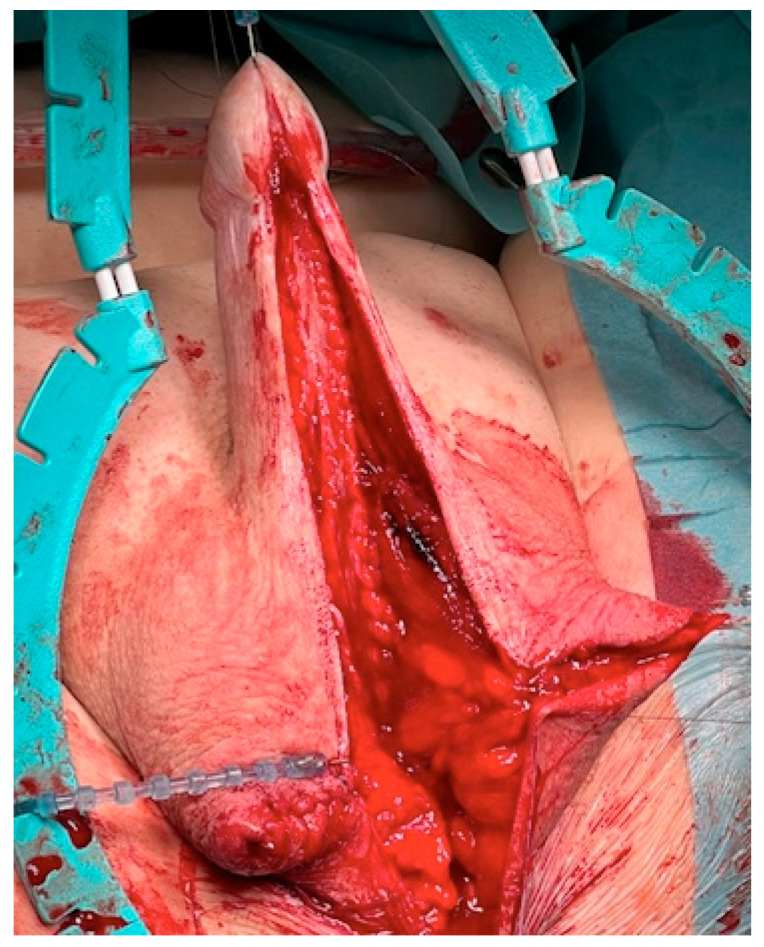
Closure of the divided raphe over the testis.

**Figure 3 jcm-11-05989-f003:**
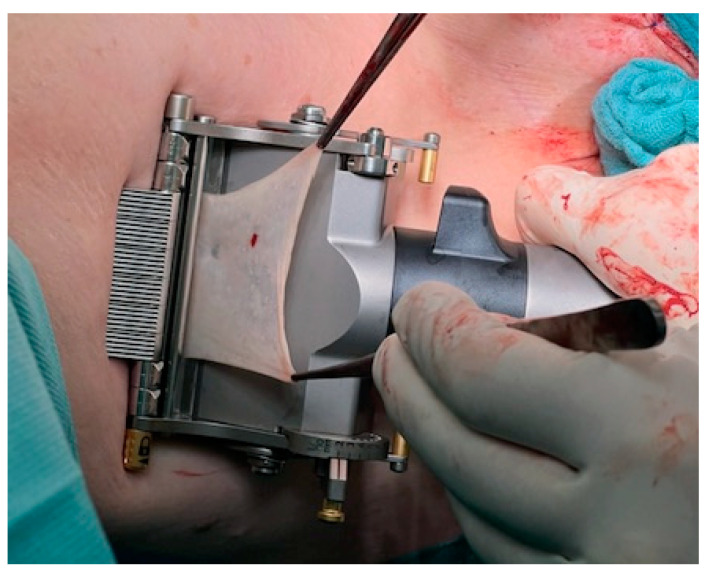
Harvesting the split-skin graft with split-skin dermatome, driven by compressed air.

**Figure 4 jcm-11-05989-f004:**
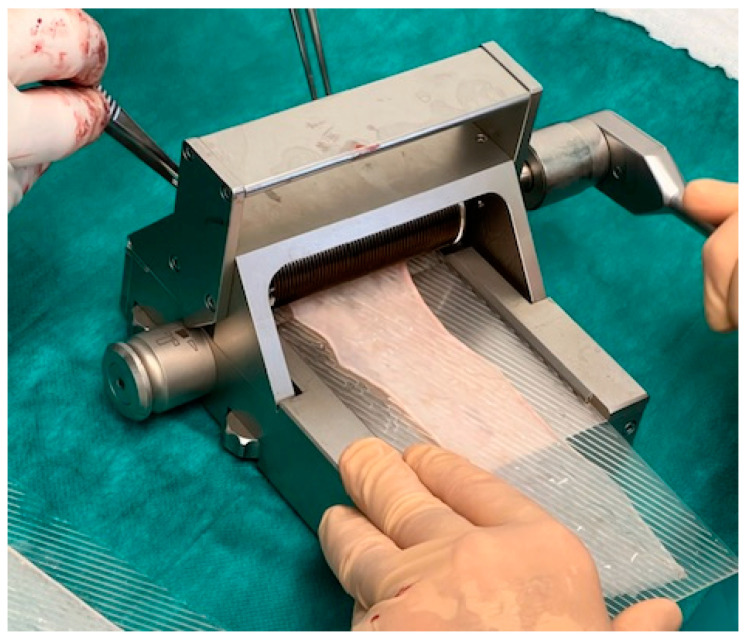
Split-skin transplant is converted into a network by a mesh graft dermatome.

**Figure 5 jcm-11-05989-f005:**
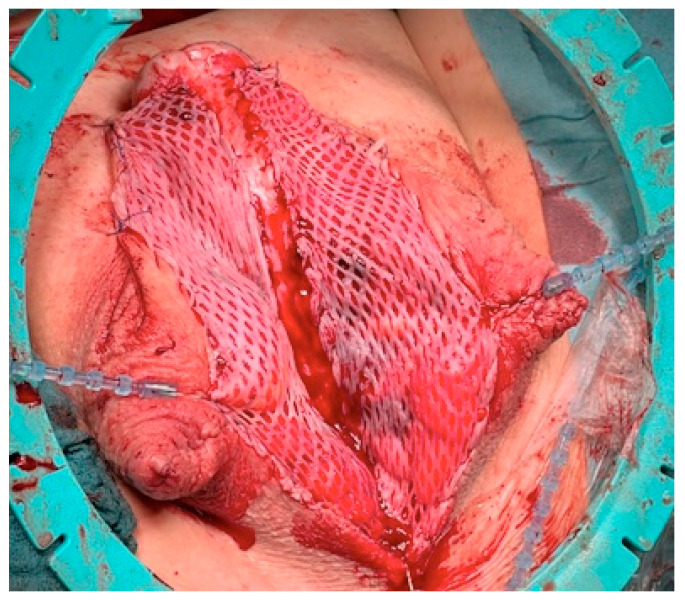
The mesh transplant is sutured between the incised urethral stricture and the skin edge.

**Figure 6 jcm-11-05989-f006:**
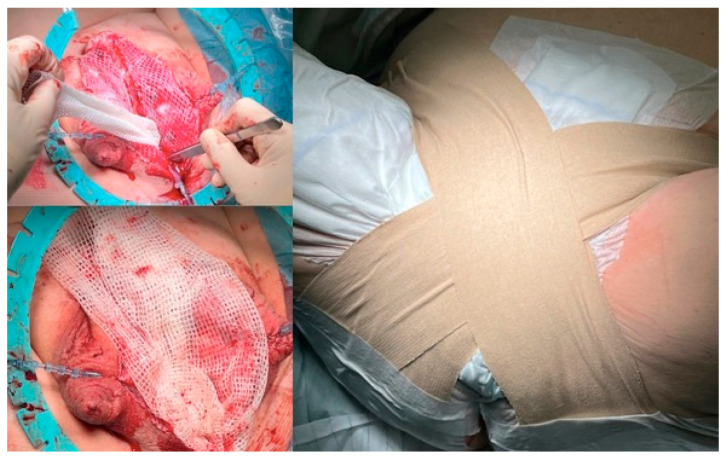
Dressing after Stage 1 Surgery.

**Figure 7 jcm-11-05989-f007:**
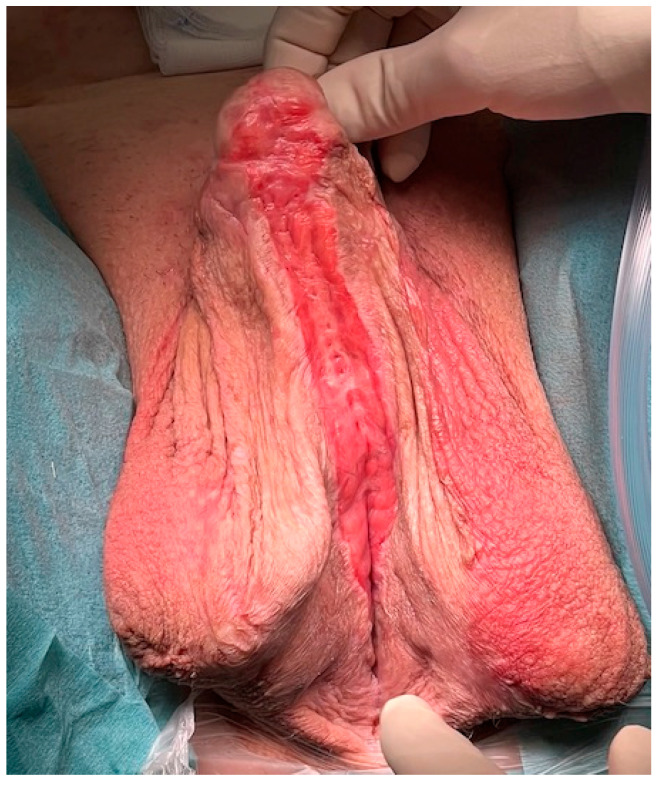
Completely healed skin transplant ensuring optimal conditions for the reconstruction of the neourethra.

**Figure 8 jcm-11-05989-f008:**
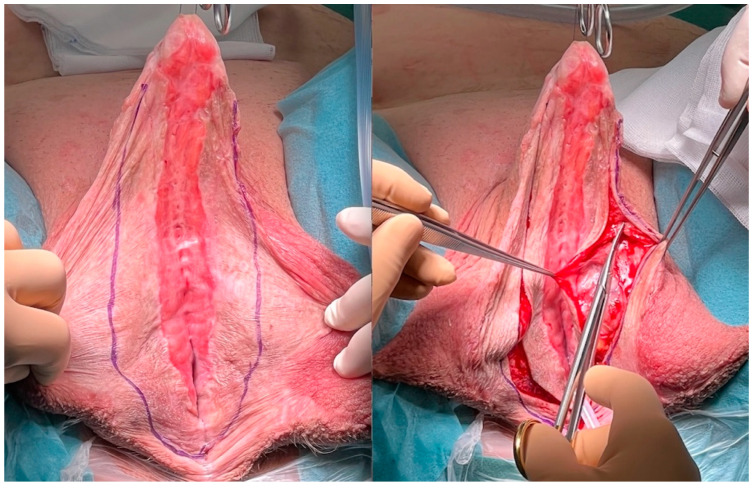
Incising and mobilising the transplanted skin.

**Figure 9 jcm-11-05989-f009:**
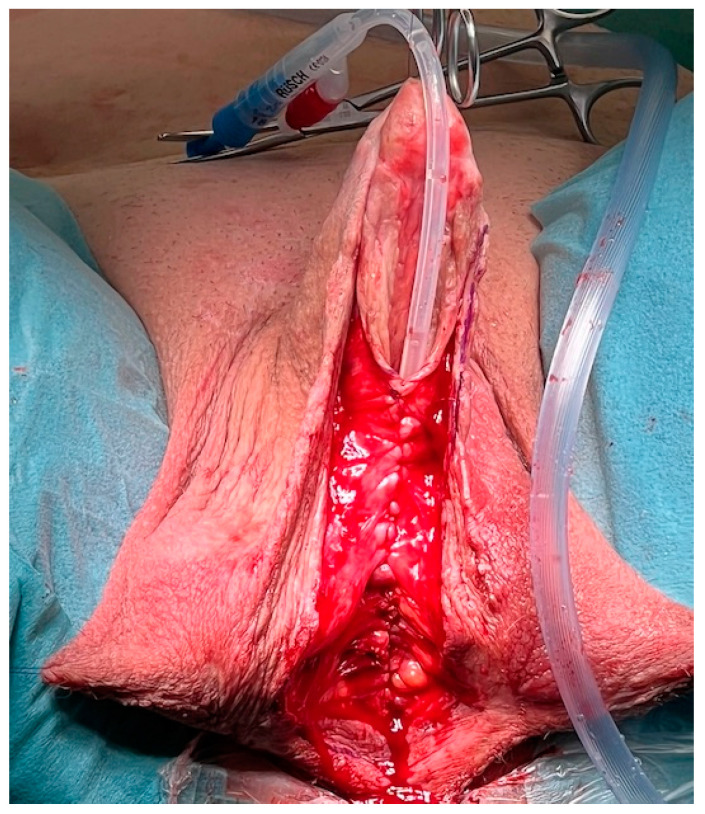
Closure of the neourethra with inverted running sutures.

**Figure 10 jcm-11-05989-f010:**
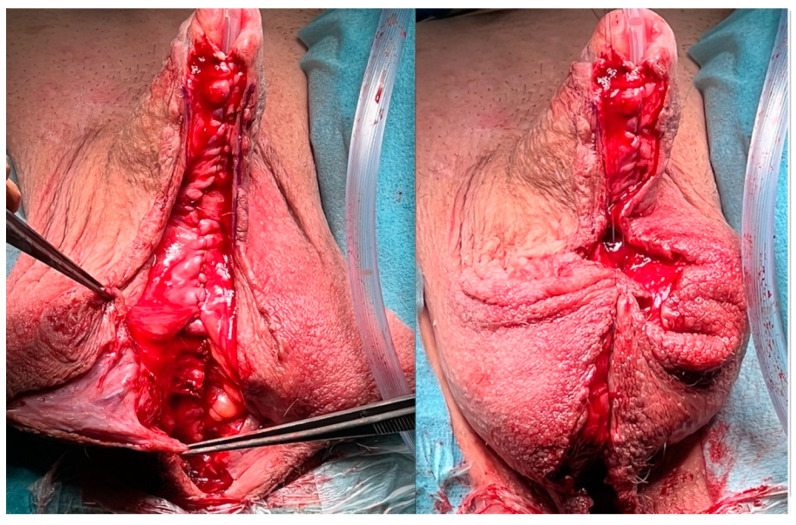
Remodeling the normal anatomy by reproducing a raphe.

## Data Availability

Not applicable.

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
