# Peer review of "Mesh Graft Urethroplasty—Still a Safe and Promising Technique in Mostly Unpromising Complex Urethral Strictures"

_jcm, 2022, doi:10.3390/jcm11205989_

Round 1

Reviewer 1 Report

I dont understand why the authors described this technique as a review article, while in discussion they provide excelent results of this techique. Also, they subjectively created their own guidelines, while EUA clearly states what is a complex or "long" stricture. This is really dissapointing and these results should be compared with reccomended operation- buccal graft, while they not introduce a novel operation technique

Author Response

Dear Reviewer 1:

Thank you very much for your constructive review on our manuscript “Mesh Graft Urethroplasty – still a safe and promising technique in mostly unpromising complex urethral stricture”.

We are pleased to have the opportunity to answer your comments:

We changed the article type from “Review article” to “Comment Article”.

Definition by the EAU on urethral strictures:

  • “… multiple interventions in the past, unfavourable clinical findings such as significant spongiofibrosis or scarring that requires excision, poor quality of the urethral plate…” (EAU Guidelines, Lumen et al. 2022)

Simple “long” urethral strictures are not defined as complex urethral strictures. As we mentioned in our manuscript, there have to be different circumstances that lead to define an urethral stricture as a “complex” urethral stricture. In our opinion, we did not diverge from this definition, as we try to precisely describe the difficulties that come along with the unpromising circumstances, that led to the complex urethral stricture and the difficulties to perform a reasonable and successful solution.

The EAU Guidelines propose using staged urethroplasty in above mentioned cases and do not prefer any transplant type. So, according to the EAU Guidelines, our therapy regime fit to theirs.

We actually agree to the existing guidelines on definition and therapy regimes on “complex” urethral strictures and discussed this issue.

We also use mesh graft in panurethral strictures, especially when co-circumstances, like the mentioned above, are present.

There are no comparative studies regarding BM and Mesh Graft. As mentioned in EAU Guidelines “it is …difficult to draw meaningful conclusions from the little data that are available…” (EAU Guidelines, Lumen et al. 2022) in regard of comparing free graft transplantation techniques.

We hope we could clarify the points you mentioned. We thank you very much for improving the quality of our manuscript and making it worth publishing.

Reviewer 2 Report

Congratulations to the authors for presenting the surgical technique and the results in such complex and difficult cases.

The information contained is really useful and interesting for those involved in reconstructive surgery.

Some aspects to consider for the paper thus presented:

to define a review it is necessary to use PRISMA criteria and to change the manuscript structure accordingly... the present form is more similar to single center experience and "narrative" review of the literature but it is not a review article

- Line 41: Please better specify spongiofibrosis classification (Grade? How to define it?)

- Line 187: Please add figure of the second stage and at the end of the second stage procedure

Discussion should be modified: 

     - reduce the paragraph about stricture imaging and diagnosis

    - improve direct comparison among the MG reconstruction you presented  with previous published series (both single and staged procedures... also cite Kulkarny panurethral reconstruction and compare it to your technique) - a resuming table could help in doing this

Author Response

Dear Reviewer 2:

Thank you very much for your constructive review on our manuscript “Mesh Graft Urethroplasty – still a safe and promising technique in mostly unpromising complex urethral stricture”.

We are pleased to have the opportunity to answer your comments:

Ad1:

We changed the article type from “Review article” to “Comment Article”

Ad 2:

We classified the spongiofibrosis. (Line 40-46).

Ad 3:

We added figures of the 2nd stage procedure.

Ad 4:

We highly believe, that the definition and diagnosing of spongiofibrosis is one of the most demanding and critical issues to choose the right surgical strategy and to ensure a successful therapy. Therefore, we believe this topic should be as precise as possible.

Ad5:

There are no comparing studies of the different surgical techniques. We tried to compare, by actual literature, the different free graft techniques and their outcome, but this is almost impossible like mentioned in the actual EAU Guidelines:  “it is …difficult to draw meaningful conclusions from the little data that are available…” (EAU Guidelines, Lumen et al. 2022).

We highlighted the limiting factors for BM surgery. We believe, that MG urethroplasty and BM urethroplasty – even in panurethral (not complex!) strictures – complement each other and should not compete. As mentioned above, we changed the manuscript from “Review Article” into “Communication Article”.

We hope we could clarify the points you mentioned. We thank you very much for improving the quality of our manuscript and making it worth publishing.

Reviewer 3 Report

In the review entitled "Mesh Graft Urethroplasty: still a safe and promising technique in mostly unpromising complex urethral strictures", the mesh graft urethroplasty surgical technique is discussed in cases of complex urethral strictures. This article attempts to bring the procedure into the present day and emphasizes the importance of continuing education and improvements for the urological arsenal.

I would like to point out that the manuscript is well structured and meets all the requirements for a review article. The surgical technique is displayed in a professional manner, being accompanied by all the steps as well as the postoperative care in every step of the procedure. Furthermore, pictures of the procedure are included so that the procedure can be supported.

The Discussions adhere to the literature and are backed up by a satisfactory number of references that are currently in the literature. In keeping with the title, the conclusions have been reached.

As a result, this review provides a professional and elegant approach to aligning the review with the specialized literature. 

Considering its suitability for publication, this manuscript is a valuable addition to the treatment arsenal for the treatment of complex urethral strictures 

Author Response

Dear Reviewer 3, 

Thank you very much for your kind words and encouraging this paper to be published. 

Reviewer 4 Report

Dear Authors, 

I would suggest to use the term panurethral stricture, since the technique that you kindly describe is applied to this type of stricture disease.

In addition, some editing is required to improve the readability of this draft. 

In the discussion section, you kindly report the initial results of MG urethroplasty in 38 men. I suggest moving this data from the discussion and create a Result section. 

Finally, the paragraph on diagnosis (RUG and VCUG) is extensive and i believe it is beyond the scope of this review. 

Author Response

Dear Reviewer 4:

Thank you very much for your constructive review on our manuscript “Mesh Graft Urethroplasty – still a safe and promising technique in mostly unpromising complex urethral stricture”.

We are pleased to have the opportunity to answer your comments:

Ad 1:

Like the EAU Guidelines, we do not believe, that panurethral strictures should be equated to complex urethral strictures.

As described in the manuscript there have to be different (complex) circumstances that lead to the definition of complex urethral strictures.

The EAU Guidelines classify complex urethral strictures as followed:

  • “… multiple interventions in the past, unfavourable clinical findings such as significant spongiofibrosis or scarring that requires excision, poor quality of the urethral plate…” (EAU Guidelines, Lumen et al. 2022)

Simple “long” or “panurethral” urethral strictures are not defined as complex urethral strictures.

In our opinion, we did not diverge from the EAU definition, as we try to describe the difficulties that come along with the unpromising circumstances that led to the urethral stricture and the difficulties to perform a reasonable and successful solution.

Ad 2:

We changed the article type from “Review article” to “Comment Article”.

Ad 3:

We changed the manuscript type form “Review” article to “Comment” article. Our results are just additionally commenting the before published results of other surgeons to proof the feasibility of the technique.

Ad 4:

We highly believe, that the definition and diagnosing of spongiofibrosis is one of the most demanding and critical issues to choose the right surgical strategy and to ensure a successful therapy. Therefore, we believe this topic should be as precise as possible.

We hope we could clarify the points you mentioned. We thank you very much for improving the quality of our manuscript and making it worth publishing.

Round 2

Reviewer 1 Report

comment article is more suitable for this kind of article

Reviewer 4 Report

The manuscript has been significantly improved